# Seasonal changes in the structure of river fish communities in temperate Japan depicted using quantitative eDNA metabarcoding

Takeshi Ito[1,2]*, Tetsu Yatsuyanagi[2], Tomohiro Yokobe[3], Misaki Shiomi[2], Reiji Masuda[2]

1 Sesoko Station, Tropical Biosphere Research Center, University of the Ryukyus, Sesoko, Motobu, Okinawa, Japan, 2 Maizuru Fisheries Research Station, Field Science Education and Research Center, Kyoto University, Nagahama, Maizuru, Kyoto, Japan, 3 Field Science Education and Research Center, Kyoto University, Oiwake-cho, Kitashirakawa, Sakyo-ku, Kyoto, Japan

* takeshi.itos12d418e@gmail.com

## Abstract

Understanding fish communities contributes to various fields ranging from theoretical ecology to fisheries management. In rivers flowing from mountainous to urban areas and showing seasonal variations in water temperatures, fish community structure may be spatiotemporally affected by natural and anthropogenic factors. We investigated the spatial and seasonal dynamics of fish communities in the Isazu River (17.9 km) and its tributary, Ikeuchi River, in temperate Japan, using eDNA metabarcoding method. We detected 78 fish operational taxonomic units across four seasons at 12 sites along the river length. The fish community differed significantly among different seasons and sites, aligning with the distance from the river mouth. In particular, the eDNA concentrations (copies/L) of major fishery resources and endangered fish species varied among different seasons and sites, which may reflect species migration patterns. The river hosts a spatiotemporally dynamic fish community that provides substantial ecosystem services. Our study provides valuable insights into the complex relationships between fish communities and natural, anthropogenic, temporal, and spatial factors affecting these communities based on changes in eDNA concentrations.

## Introduction

Freshwater biodiversity and ecosystems are threatened by anthropogenic activities and climate change [1]. Fish are the most diverse animal taxa among vertebrates, and understanding fish community structure is important for the conservation of endangered species and for the management of fishery resources, provision of ecosystem services, and understanding of freshwater biodiversity. Therefore, a comprehensive understanding of fish communities is essential for ensuring sustainable ecosystem services. Freshwater fish are often endemic to a region [2,3], and

**Data availability statement:** The dataset and R script used in this study are publicly available on Dryad at [https://doi.org/10.5061/dryad.tx95x6b88].

**Funding:** This project was funded by the AEON Environmental Foundation to RM and Japan Society for the Promotion of Science KAKENHI (grant number: 19H05641 to RM). The funders had no role in study design, data collection and analysis, decision to publish, or preparation of the manuscript.

**Competing interests:** The authors have declared that no competing interests exist.

the structure of the fish community differs among rivers, even in neighbourhoods and tributaries [4–9] likely because of the impact of multiple natural factors, including the geographical features of a river, and anthropogenic factors, such as the presence of artificial dams and water pollution [10].

Natural factors reflect the geographical characteristics of the area. River topography impacts fish communities, and fish biodiversity and species composition may vary from upstream to downstream along the river length [11–15]. River size is a major factor in determining fish communities [16,17], and small rivers are considered relatively more unstable and sensitive to a variety of factors, such as changes in water temperature and quality, than larger rivers. In addition, the slope of the river affects the migration of diadromous fish between the sea and rivers [18]. Climate variability throughout the year is also affected by the geography of the area. In temperate regions, such as those having humid subtropical climates with four seasons, changes in river water temperature affect fish population dynamics [19–21]. Differences in water temperature affect the seasonally dependent migrations of fish species and impact the patterns of change in fish communities [6,22,23].

In addition, anthropogenic factors impact fish communities. The impact of anthropogenic factors on fish communities depends on the urbanisation of the area through which the river flows [24]. In highly urbanised areas, river water contamination with sewage reduces macroinvertebrate biodiversity [25]. In addition, artificial dams limit fish migration and affect fish communities [26]. On the contrary, the proper management of ecosystems can secure fishery resources, ecosystem services, and societal values [27–29]. Therefore, understanding fish community structure and its interaction with environmental or spatial factors, including anthropogenic factors, is necessary to guide efforts to ensure the provision of sustainable ecosystem services.

As fish community structure is temporally and spatially affected by natural and anthropogenic factors, sampling at multiple sites over a long period is desirable to obtain high-resolution data on fish communities [14]. Analyzing environmental DNA (eDNA), which is DNA present in the environment, may be useful to assess spatio-temporal variations in fish community structure. eDNA is the genetic material emitted by organisms that can be detected in the environment [30]. It has been used to identify and detect aquatic organisms, particularly fishes, amphibians, and aquatic insects [31–35], and to conserve and monitor endangered species [36–38]. eDNA-based analysis has now become cost-efficient and has been utilised to estimate the structure of fish communities on a massive scale [39,40]. In particular, the eDNA metabarcoding approach, which can identify multiple biotas in parallel using taxon-universal polymerase chain reaction (PCR) primers, is a powerful tool for analysing fish communities [6,8,41,42].

In this study, we investigated riverine fish communities using eDNA metabarcoding to identify fish taxa in the Isazu River Basin in Maizuru, Japan. The Isazu River is 17.9 km long and flows to the Maizuru Bay through the small city of Maizuru (a population of approximately 80,000 people), and the Ikeuchi River joins the middle reaches of the Isazu River. Thus, this river is exposed to anthropogenic activity but less than the rivers in large cities. In addition, there are two small weirs, one on each

river. The weir on the Isazu River has a height of 1.5 m and that on the Ikeuchi River has a height of 10 m and has an outlet at a height of 2.2 m (see Materials and Methods). Herein, we aimed to examine how natural and anthropogenic factors affect ecosystem services, such as diversity in freshwater fish communities, including endangered fish species, and the supply of fishery resources. The following questions were addressed: (1) How do fish communities undergo seasonal changes in temperate river systems, where water temperatures vary greatly with the season? (2) Do major fishery resources and endangered fish species exhibit seasonal migration patterns? (3) How do fish communities differ from upstream to downstream along the river length, and how do they differ between the mainstream and the tributary? (4) Is the fish community affected by water quality (i.e., indicators of human activity)? (5) Is fish migration restricted by weirs? The results obtained in this study enhance our understanding of the ecological dynamics of river fish communities and highlight the importance of small local rivers in maintaining biodiversity and supporting fishery resources.

## Materials and methods

### Study sites and eDNA samplings

We conducted eDNA sampling in the Isazu River and its tributary, the Ikeuchi River, in Maizuru, Kyoto, Japan. Twelve sites (Sts. 1–12) were established as eDNA sampling locations (Fig 1a, S1 Table). The Isazu River is lined with some form of bank protection along nearly all its banks, except at St. 12, to prevent flooding. However, sediment has accumulated inside the riverbanks, creating a non-linear terrain within the river. The elevation of the uppermost site (St. 12) is 213 m, and the tributary, the Ikeuchi River, is shorter and steeper than the mainstream Isazu River (Fig 1b). The samplings were conducted every 3 months from May 2023 to February 2024 (S1 Table). Herein, we assigned May as spring, August

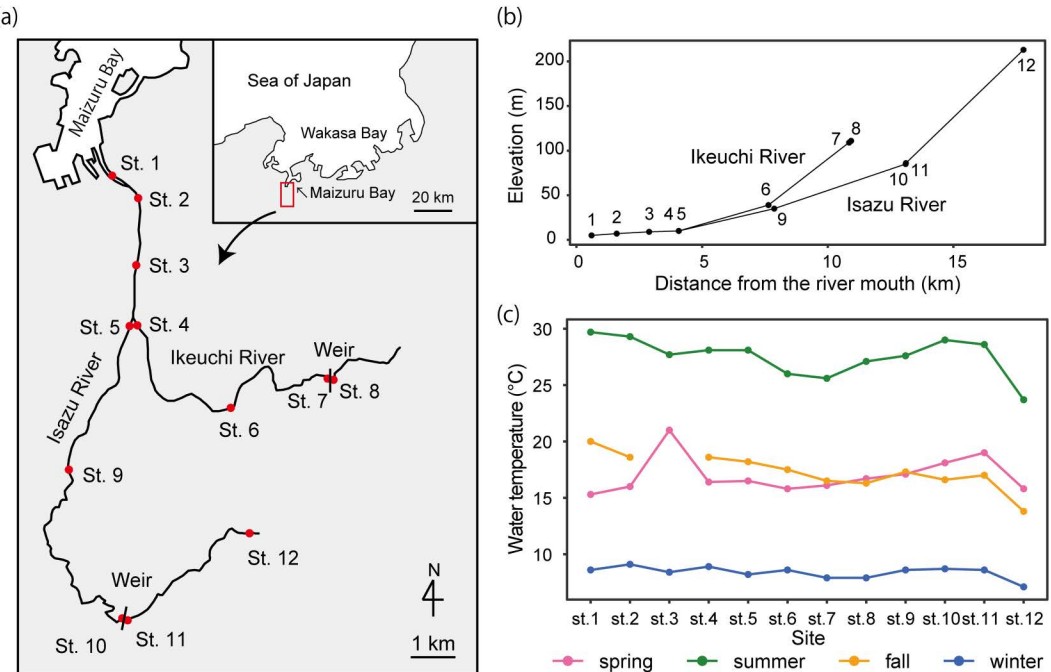

**Fig 1. Map showing the water sampling sites used in this study.** (a) Map of the Isazu basin. The tributary stream, i.e., the Ikeuchi River, joins in the middle reaches of the Isuzu River. There is one small weir between Sts. 7 and 8 and another between Sts. 10 and 11. The map is created by editing the digital GSI map published by the Geospatial Information Authority of Japan (Ministry of Land, Infrastructure, Transport and Tourism, https://maps.gsi.go.jp/). The figure is similar but not identical to the original image and is therefore for illustrative purposes only. (b) Relationships between distance from the river mouth (km) and elevation (m) at each site. (c) Water temperatures at each site during different seasons.

as summer, November as fall, and February as winter, following the Japan Meteorological Agency. Water temperatures varied among seasons; however, the water temperature ranges were similar in spring and fall (Fig 1c). In each season, the water temperatures were comparatively consistent at Sts. 1–11, whereas the water temperature was relatively lower at St. 12 (Fig 1c). The Ikeuchi (height: 10 m) and Isazu (1.5 m) Rivers have one weir each, and the weir on the Ikeuchi River has an outlet located at a height of 2.2 m. In both weirs, there were no fish ladders. Samplings below and above the weirs were conducted at Sts. 7 and 10 and Sts. 8 and 11, respectively (Fig 1a). Note that we did not sample water at St. 3 in November because of ongoing construction work on the river in that period.

Water samples (5 L) were collected from each site using a bucket. Of which, 500 mL was filtered at each site using a Sterivex™ cartridge (Merck Millipore, Darmstadt, Germany) with a filter pore size of 0.45 µm, and this procedure was repeated twice, resulting in a total filtration of 1 L. After filtering the water sample, 2 mL of RNAlater™ (Thermo Fisher Scientific, Waltham, MA, USA) was added to the cartridge for DNA preservation. A negative control was prepared by filtering 500 mL of distilled $H_2O$ at the end of the day. All samples were stored at −20°C until DNA extraction.

## eDNA metabarcoding and bioinformatic processing of eDNA sequence data

Total DNA was extracted using a DNeasy Blood and Tissue DNA Extraction Kit (Qiagen, Hilden, Germany) [43]. RNAlater in the cartridge was drained by centrifugation at 5,000 × $g$ for 2 min. Approximately 20 µL of protease K mixed with 400 µL of Buffer AL was added to the cartridge, and then the cartridge was incubated at 56°C for 30 min with rotation. Subsequently, the solution containing DNA in the cartridge was retrieved by centrifugation, and 200 µL of ethanol was added to this solution. Next, we filtered and purified the DNA solution using a spin column, following the manufacturer's instructions. These procedures were performed on two filtered cartridges per site. The final volume of extracted DNA was 100 µL per filter cartridge, and 10 µL of extract from two cartridges was combined to serve as the PCR template per site.

All work involving eDNA metabarcoding, including library preparation, Illumina MiSeq sequencing, and bioinformatic processing of eDNA sequence data, was outsourced to Environmental Research & Solutions Co. Ltd. (Kyoto, Japan, http://www.ctiers.co.jp/). Our library preparation protocol was based on the workflow of quantitative MiFish metabarcoding [44]. The first-round PCR (1st PCR) was performed on eight replicates of each sample using the fish universal primer set MiFish-U-F/R [45]. The sequences of the MiFish primers are as follows: NNNNNNGTCGGTAAAACTCGTGCCAGC (MiFish-U-F) and NNNNNNCATAGTGGGGTATCTAATCCCAGTTTG (MiFish-U-R). The reaction mix contained 6.0 µL of 2 × KAPA HiFi HotStart ReadyMix (KAPA Biosystems, Massachusetts, ILM, USA), 0.7 µL of each primer (5 µM), 1.6 µL of sterilised distilled $H_2O$, 1.0 µL of internal standard DNA mix, and 1.0 µL of template, and the final volume of the reaction mix was made up to 12 µL. The internal standard DNA solution containing the artificially synthesised DNA of non-endemic fish species in Japan (*Acanthopsoides gracilentus*, *Elopichthys bambusa*, and *Labeo coubie* at 5, 25, and 50 copies/µL, respectively) was added to the reaction solution to estimate the number of copies of fish DNA sequences. The thermal cycle profile in the 1st PCR was as follows: 1 cycle of 3 min at 95°C; 38 cycles of 20 s at 98°C, 15 s at 65°C, and 15 s at 72°C; and 1 cycle of 5 min at 72°C. The PCR products were purified using AMPure XP magnetic beads (Beckman Coulter, Brea, CA, USA), and the resulting DNA concentrations (ng/µL) are summarised in S2 Table. Subsequently, the second-round PCR (2nd PCR) was performed in a 12 µL reaction volume containing 6.0 µL of 2 × KAPA HiFi HotStart ReadyMix, 0.83 µL of each primer (5 µM), 2.34 µL of sterilised distilled $H_2O$, and 2.0 µL of the PCR product from the 1st PCR diluted to 0.1 ng/µL. The thermal cycle profile in the 2nd PCR was as follows: 1 cycle of 3 min at 95°C, 12 cycles of 20 s at 98°C and 15 s at 72°C, and 1 cycle of 5 min at 72°C. Non-specific amplification products were eliminated using an E-Gel system (Thermo Fisher Scientific, Waltham, MA, USA) for size selection, and the size of the amplification products was measured using a BioAnalyzer (Agilent Technologies, Santa Clara, CA, USA) to confirm whether the target sequence was obtained. The concentration of the products was measured using a QuantiFluor ONE dsDNA System (Promega, Madison, WI, USA), and the concentration of the libraries was measured using the sequence length and molecular weight of 1 bp

(660 g/mol). After adjusting the library concentration to 11 pmol/L, sequencing was performed on the MiSeq platform using the MiSeq v2 Reagent Kit for 2 × 150 bp PE (Illumina, San Diego, CA, USA).

MiSeq raw reads were processed (trimmed, merged, quality-filtered, and denoised) using the PMiFish pipeline version 2.4.1 [42] (https://github.com/rogotoh/PMiFish) and USEARCH version 11 [46]. Amplicon sequence variants obtained through read processing were subjected to taxon assignments for being assigned to different operational taxonomic units (OTUs) using the usearch_global command with a sequence identity of >98.5% with reference sequences. MiFish DB (version 2024-04-16) was used as the reference database. The DNA copy numbers of each OTU were calculated based on the read numbers of the internal standard DNAs. The detected read numbers of them were shown in S3 Table, and the coefficient of determination ($R^2$) calculated from the detected reads of internal standard DNA and the predefined copy numbers were 0.93–1.00. The copy numbers per μL of the PCR template was multiplied by 200 to convert it to the copy numbers per L of environmental water, based on the total volume of the extracted solution (20 μL used for the analysis × 10 to account for the total 200 μL extract, which represents the entire filtration of 1 L). When multiple species shared the same aligned sequence, they were combined at the genus or family levels. Detection sequences with copy numbers ≤1 per 1 L of environmental water were excluded from the dataset (nonapplicable in this study). We eliminated the sequences of non-Japanese species that were probably obtained through contamination from domestic wastewater or were derived from sequencing errors.

## Measurement of water quality index at each site

The concentrations of several dissolved substances (indicators of anthropogenic activities) were measured at each site to determine the impact of these substances on the fish community. Approximately 50 mL of river water was filtered through a hydrophilic PTFE membrane filter with a pore size of 0.45 μm (Advantec, Taipei, Taiwan). Water was analysed using a QuAAtro2-HR autoanalyzer (BLTEC, Osaka, Japan). The measurement variables were nitrite nitrogen ($NO_2$-N, mg/L), nitrate nitrogen ($NO_3$-N, mg/L), and orthophosphate phosphorus ($PO_4$-P, mg/L) concentrations. In addition, we measured water temperature (°C) and salinity (ppt) and estimated river width (m), elevation (m), and the distance from the river mouth (km) for each site.

## Data analyses

All analyses were performed using the R software version 4.4.0. Alpha diversity was calculated using the Shannon diversity index H′ using the R package 'vegan' [47], and the index was subsequently exponentiated to restore its doubling property and express diversity as the effective number of species [48]. To assess whether the Shannon diversity index (H′) varied with season and distance from the river mouth, we fitted a generalized linear model (GLM) including season, distance, and their interaction as explanatory variables. The model was specified with a Gamma distribution and a log-link function. Likelihood ratio tests using Type II sums of squares were conducted with the R package 'car' [49]. To detect the species structure of the fish community, we performed a non-metric multidimensional scaling (NMDS) analysis based on the Bray–Curtis similarity index using the R package 'vegan'. In addition, we performed the permutational multivariate analysis of variance (PERMANOVA) to test the differences in similarity in community structure among different seasons or sites, followed by a pairwise test based on the Bray–Curtis index using the R package 'pairwiseAdonis' [50]. The number of permutations was set to 999. Environment data that significantly affected the community ($p < 0.05$) was plotted on the NMDS plot using 'vegan'. In addition, we performed cluster analysis using Ward's method for clustering data, and the optimal number of clusters was determined by silhouette analysis using the R packages 'cluster' and 'dendextend' [51]. These analyses were performed for each season and for the entire year. In addition, as the distance from the river mouth and the concentrations of several dissolved substances were significantly related to the fish community, we constructed GLMs with water quality variables as the response and season, distance, and their interaction as predictors. These models also used a Gamma distribution with a log-link

function, followed by likelihood ratio tests with Type II sums of squares. As the data for $NO_2$-N and $PO_4$-P concentrations contained zero values, we added 0.0001 to these variables. As the GLM using $PO_4$-P concentration data did not converge under the default settings (25 iterations), the number of iterations was set to 100. The outliers of $NO_2$-N concentration data in fall were excluded from the analysis because of their disproportionate impact on the model.

In addition, we investigated seasonal changes in the eDNA concentrations of major fishery resources (Japanese eel: *Anguilla japonica*; and ayu sweetfish: *Plecoglossus altivelis altivelis*) and endangered fish species (Japanese eel: *A. japonica*; fluvial eight-barbel loach: *Lefua torrentis*; and Japanese torrent catfish: *Liobagrus reinii*) at each site using the R package 'superheat' [52]. Note that Japanese eel is an endangered species while also being a major fishery resource.

Furthermore, we investigated whether the weirs act as a barrier to fish migration. The distance between St. 7 and 8 is about 100 m and between St 10 and 11 is about 10 m. Therefore, eDNA from upstream of the weirs may be detected downstream. However, if the weirs act as a barrier to fish movement, certain species should only be detected below them. We listed the species that appeared at least once upstream or downstream of the weirs in all seasons and examined whether there were any species found only downstream.

### Ethics statement

We conducted field research. All sampling sites were public locations, and since only a small amount of river water was collected, no permission was required in Japan.

## Results

### Summary of fish taxa and biodiversity

We detected 122 fish OTUs throughout the year, including 17 identified at the genus level and 1 at the family level (S4 Table). In spring, summer, and fall, no eDNA was detected in any of the negative controls, whereas three OTUs (2 copies/L of *Rhinogobius flumineus*, 70 copies/L of *Cyprinus carpio*, and 156 copies/L of *Pseudogobio esocinus*) were detected in the negative control in winter. Therefore, we subtracted these values from the data of these OTUs obtained for each site. After excluding OTUs presumed to originate from drained water and marine fish, except for diadromous fish and brackish water species that could be detected in the river mouth, a total of 78 OTUs, including 11 identified at the genus level, were ultimately detected. Near St. 1–3, there are fishing ports and fish markets, leading to the detection of many oceanic fish. As a result, the exclusion rate was relatively high in certain seasons (S4 Table). The identified OTUs comprised 57 genera and 26 families, of which 10 were endangered species according to the Ministry of the Environment of Japan [53] and the IUCN Red List (indicated in red letters in Fig 2a), and 5 were invasive species (indicated in blue letters in Fig 2a). Twenty-one OTUs were detected in all seasons (Fig 2a). The number of specific OTUs per season were as follows: 5 in spring, 26 in summer, 9 in fall, and 0 in winter. In addition, the number of OTUs varied among different sites (Fig 2b). In spring, fall, and winter, the number of OTUs tended to be higher downstream than upstream. In summer, however, species richness was relatively lower at the river mouth and uppermost reach of the river compared to the middle reaches (Fig 2b). The highest eDNA concentration detected was of *R. flumineus*, followed by that of *Nipponocypris temminckii* (Fig 2c). At the genus and family levels, high eDNA concentrations of *Rhinogobius* (Gobiidae) and *Nipponocypris* (Cyprinidae) were detected (Figs 2d and 2e).

Exponentiated Shannon diversity index H′ correlated negatively with distance from the river mouth ($X^2 = 47.04$, $p < 0.001$; Fig 3). However, this correlation did not differ significantly across seasons ($X^2 = 3.54$, $p = 0.315$), although the interaction between season and distance from the river mouth was significant ($X^2 = 12.74$, $p = 0.005$).

### Spatio-temporal dynamics of community structure

The NMDS plots showed clear dissimilarities in fish communities among different seasons (PERMANOVA: $F_{(3, 43)} = 5.212$, $p < 0.001$; Fig 4a, Table 1). Fish community structures were significantly different between all pairs of seasons, except for

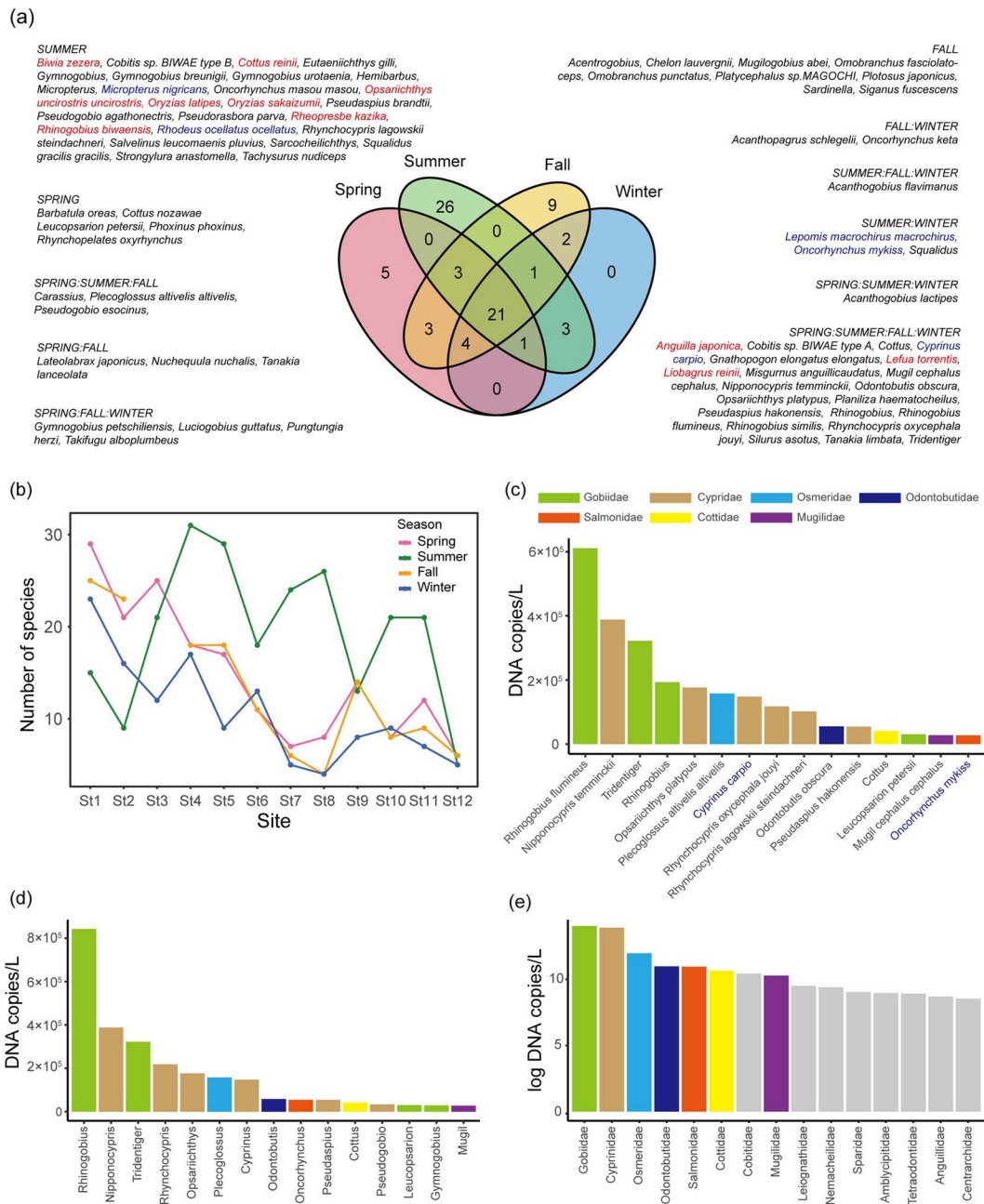

**Fig 2. Summary of the fish taxa detected in this study.** (a) Venn diagram for the four seasons. Red letters indicate endangered species according to the Ministry of the Environment of Japan and the IUCN Red List, and blue letters indicate invasive species. (b) Number of OTUs at each sampling site during the four seasons. (c) Graph showing eDNA concentrations in terms of fish eDNA copies/L, as detected in this study. (d) eDNA concentrations in terms of eDNA copies/L at the genus level. (e) eDNA concentrations in terms of eDNA copies/L at the family level. The eDNA concentrations at the family level were highly variable and, thus, are expressed in log-transformed values.

those between the spring and fall (Table 1). The structures of fish communities also differed among sites ($F_{(11, 35)}$ = 1.421, $p$ = 0.013; Fig 4b), although the differences were not statistically significant between different sites after post hoc test for each pair using the PERMANOVA pairwise method (all adjusted $p$ = 1.0).

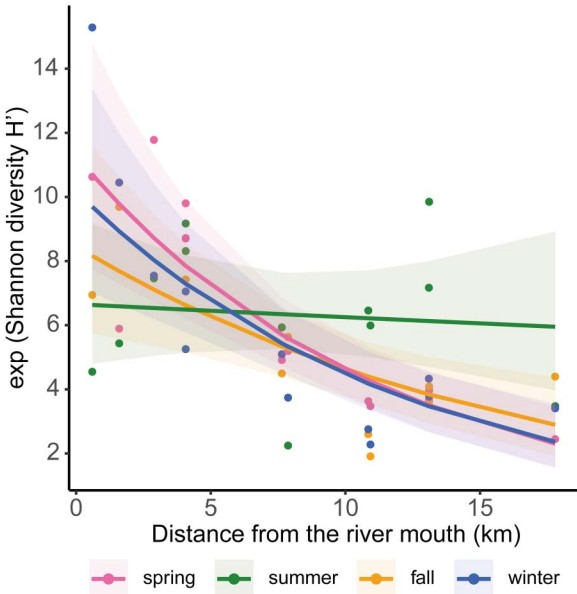

**Fig 3. Relationship between distance from the river mouth and exponentiated Shannon diversity index H′ during different seasons.**

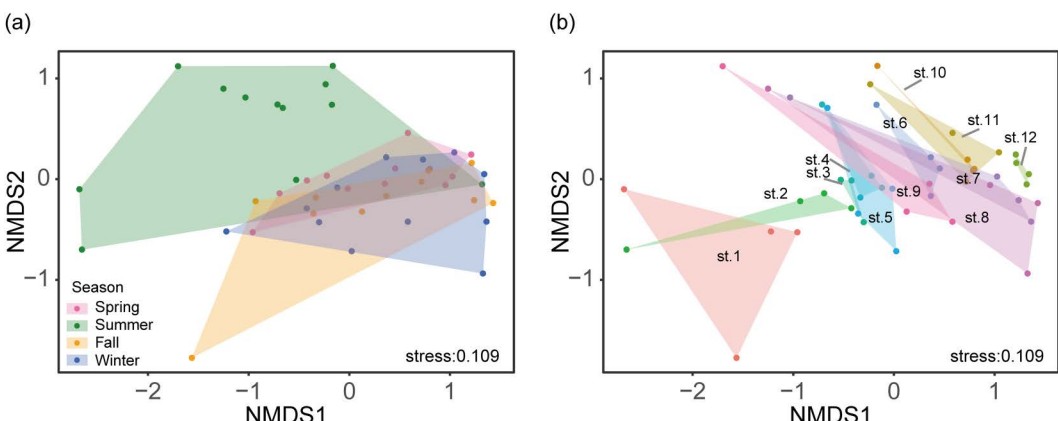

**Fig 4. Visualisation of fish communities using a non-metric multidimensional scaling (NMDS) plot.** (a) Among seasons, and (b) among sites.

**Table 1. The results of pairwise comparison of community structures among seasons, performed after the permutational multivariate analysis of variance (PERMANOVA).**

| Pairs | Df | F | $R^2$ | Adjusted-p |
|---|---|---|---|---|
| Spring vs summer | 1, 22 | 4.170 | 0.159 | 0.018 |
| Spring vs fall | 1, 21 | 1.094 | 0.050 | 1.000 |
| Spring vs winter | 1, 22 | 6.765 | 0.235 | 0.006 |
| Summer vs fall | 1, 21 | 3.623 | 0.147 | 0.018 |
| Summer vs winter | 1, 22 | 9.474 | 0.301 | 0.006 |
| Fall vs winter | 1, 21 | 6.551 | 0.238 | 0.006 |

*p*-values were adjusted by Bonferroni method. Df indicated between-group degree of freedom on left and residual degree of freedom on right, respectively.

Several environmental factors, including water quality, were associated with the fish community structure (S5 Fig). In particular, elevation and distance from the river mouth were associated with the community structure. Moreover, $NO_2$-N, $NO_3$-N, or $PO_4$-P concentrations were also associated with the fish community structure, except in the winter season (S5 Fig). The concentration of $NO_2$-N tended to be different among seasons (season: $X^2 = 38.747$, $p < 0.001$; distance: $X^2 = 1.552$, $p = 0.214$; interaction: $X^2 = 10.487$, $p = 0.015$; S6 Fig a), whereas that of $NO_3$-N was associated with season and positively associated with distance from the river mouth (season: $X^2 = 81.799$, $p < 0.001$; distance: $X^2 = 33.849$, $p < 0.001$; interaction: $X^2 = 9.334$, $p = 0.025$; S6 Fig b) and was the smallest in summer. In addition, the concentration of $PO_4$-P was associated with season and distance from the river mouth (season: $X^2 = 9.222$, $p = 0.026$; distance: $X^2 = 46.972$, $p < 0.001$; interaction: $X^2 = 3.087$, $p = 0.378$; S6 Fig c)

The structure of the fish community was divided into five clusters in spring, summer, and winter and into three clusters in fall (Fig 5). In spring, summer, and fall, sampling sites along the Ikeuchi River grouped into the same clusters as those along the Isazu River, indicating community similarity between the two rivers (Fig 5a–5c). In contrast, in winter, one cluster of the Ikeuchi River (Sts. 7 and 8) was distinct from those of the Isazu River (Sts.7 and 8). In particular, winter fish communities differed between Sts. 4 and 5, located at the root of the branch and as close as 50 m apart but belonging to different rivers (Fig 5d).

## Annual distribution of major fishery resources and endangered fish

The eDNA of the Japanese eel (*A. japonica*) was detected throughout the year, and its concentration was the highest in summer (Fig 6a). However, detections of the eels in winter were low (2 copies/L). Another major fishery resource, ayu sweetfish (*P. altivelis altivelis*), was detected in spring, summer, and fall but not in winter (Fig 6b). As for migratory species, the eDNA concentration of the Japanese eel tended to be higher upstream during summer, whereas that of ayu sweetfish was higher upstream from summer to fall. Three endangered species were detected throughout the year (Fig 2a): *A. japonica* (Fig 6a), fluvial eight-barbel loach (*L. torrentis*), and Japanese torrent catfish (*L. reinii*). In spring, *L. torrentis* was detected at all sampling sites along the river length, whereas in other seasons, it was detected only at the sites from midstream to upstream (Fig 6c). *L. reinii was* also detected through river mouth to the uppermost reach in spring and detected midstream to upstream in the other seasons (Fig 6d).

## Influence of the weirs on fish migration

A total of 34 OTUs were detected throughout all seasons in the weir between St. 7 and 8 (a height of 10 m with an outlet at a height of 2.2 m) in the Ikeuchi River. Among them, *Gymnogobius urotaenia* and *Pseudogobio agathonectris* were found only in the downstream of the weir, whereas both fish were also detected in the upstream of weirs (a height of 1.5 m) in Isazu River (St 10 and 11). In the weir between St. 10 and St. 11 of the Isazu River, 30 OTUs were detected. Among them, *Pseudorasbora parva* and *Carassius* spp. were detected only downstream of the weir, but not upstream, although *P. parva* was found exclusively at St. 7 in summer among all sites.

## Discussion

We found that the fish community structure fluctuated seasonally in the Isazu River Basin, which is consistent with the findings of Loneragan and Potter [54], who reported distinct fish fauna during different seasons in a temperate Australian estuary. However, our results contrast with the results reported for tropical streams, where no clear seasonal variation exists [55]. Water temperature is considered to be the factor that causes these differences [56]. During summers, water temperatures are at their highest ($27.5 \pm 1.7°C$) levels, which may have led to increased fish detections because of their heightened activity levels. Notably, several OTUs were identified only in the summer season, and the number of OTUs at each site in summer showed a distinct trend compared with that in the other seasons. Conversely, the lowest number of detections were observed in winter, likely reflecting the restricted activity in addition to the lower abundance of fish species

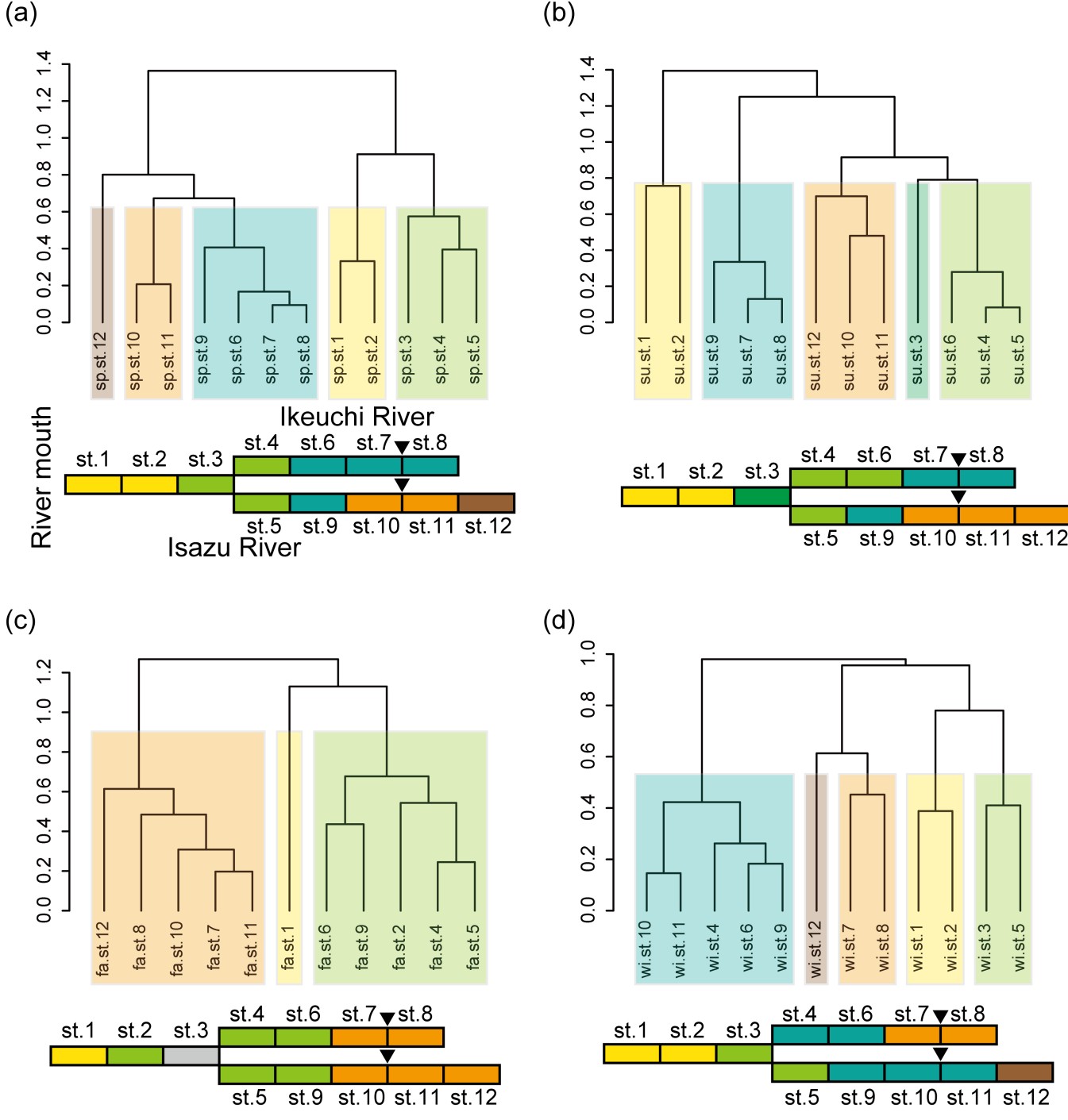

**Fig 5. Cluster dendrogram of fish communities in each season.** (a) Spring, (b) summer, (c) fall, and (d) winter. An optimal number of clusters was determined by the silhouette analysis. The illustration under the cluster indicates a schematic diagram of the Isazu River basin. The colours of the boxes in the illustration correspond to the colours of the cluster. Black arrowheads indicate the location of the weirs. St. 3 in fall is marked in grey since water could not be sampled.

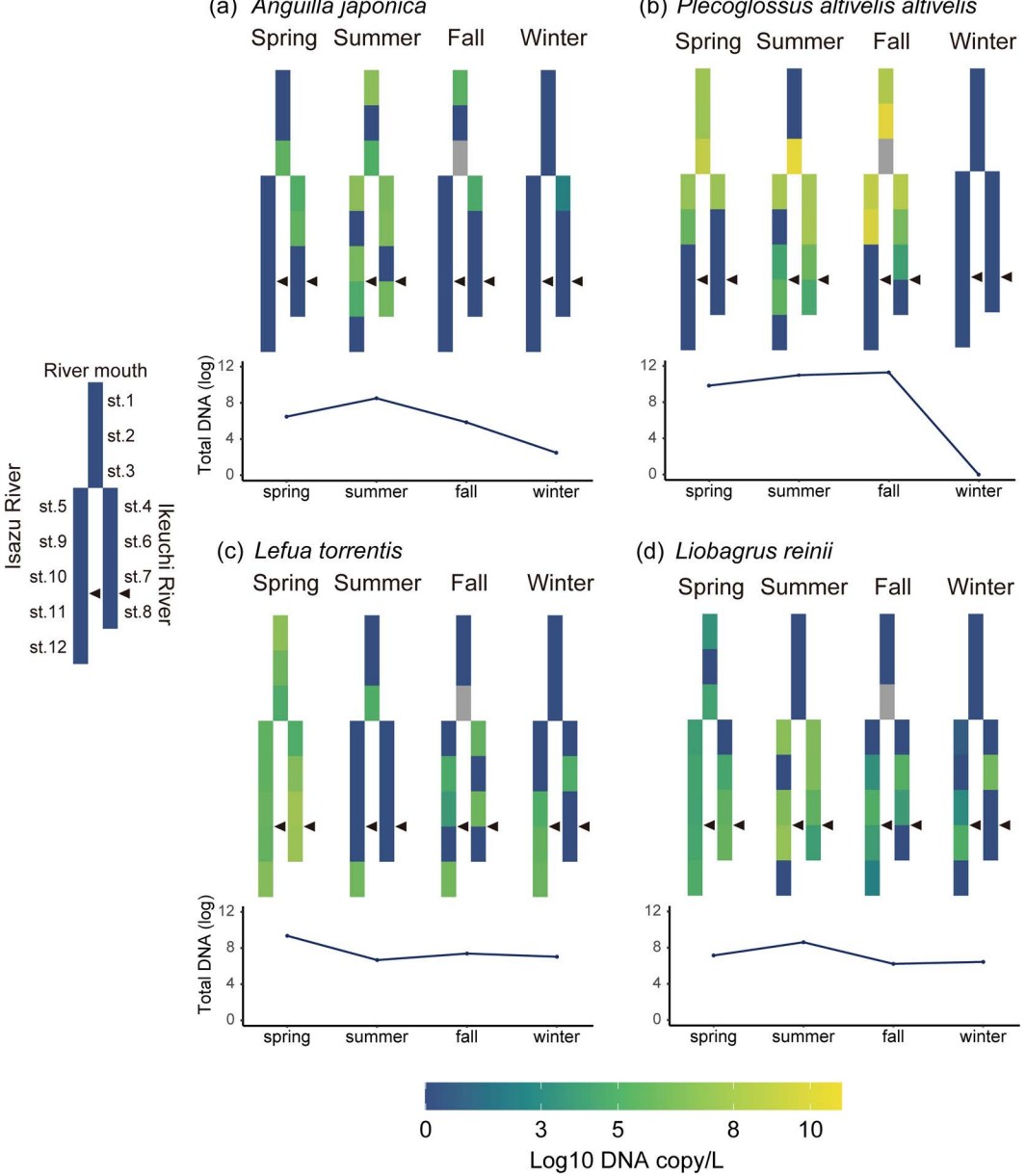

**Fig 6. Heatmap of eDNA concentration at each site on the schematic diagram of Isuzu River.** (a) *Anguilla japonica*, (b) *Plecoglossus altivelis altivelis*, (c) *Lefua torrentis*, and (d) *Liobagrus reinii*. Black arrowheads indicate the location of the weirs. The total amount of DNA (log copies/L) in each season was shown under the schematic diagram. St. 3 in fall is marked in grey since water could not be sampled.

during the coldest period (8.4±0.5°C) than that in the other seasons. In spring and fall, no significant differences were observed in the fish community structure (note that the community structure at only St. 1 in fall was an outlier). The average water temperature along the river length was similar between spring (17.0±1.6°C) and fall (17.3±1.6°C). However, it is important to note that these data represent a single day of sampling per season, providing only a rough sketch. In other rivers, the annual variation patterns in species numbers differ depending on the river type, and the number of species is not necessarily higher in summer than that in other seasons [6].

In addition, the fish communities were spatially diverse and depended on their distance from the river mouth. Moreover, on average, alpha diversity was negatively associated with the distance from the river mouth, except in summer. Usually, the upstream or headwaters are steeper and can hold relatively less water, whereas the lower stream can hold a larger volume of water [57]. In addition, nutrients can be supplied to the lower reaches from the upper reaches [58]. These differences in complicated hydro morphological conditions are among the factors that cause differences in fish assemblages. In particular, the river mouth (St. 1) appeared to have different fish communities in the NMDS plot, and the alpha diversity at this site was higher than that at the other sites. The river mouth is a place where freshwater and seawater mix and provides foraging, breeding, and habitats for multiple fishes [59]. These distinct characteristics of this site may have resulted in the differences in fish communities between this site and the other sites.

In contrast, eDNA may also be supplied from upstream, leading to a potential discrepancy between the actual presence of fish and the detected amount of eDNA. The detectability of eDNA in downstream is influenced by factors such as water flow, amount of eDNA in upper stream, and temperature, making it difficult to definitively determine at what distance upstream influences can be ruled out [60]. According to *The eDNA Society's Environmental DNA Sampling and Experiment Manual*, the distance over which environmental DNA reflects biological distribution is estimated to be around several hundred meters [43]. At least, even for *Rhinogobius flumineus*, which had the highest eDNA detection across the entire river, there were instances where it was detected immediately upstream of a certain site but not at that site itself, yet it was detected further downstream. This pattern was also observed in other species (e.g., see Fig 6).These patterns suggest that, in this river, the distance between sampling sites except for those upstream and downstream of the weirs, was likely sufficient for upstream-derived eDNA to be diluted below the detection threshold.

Fish communities did not differ between the upper mainstem of the Isazu River and its tributary (the Ikeuchi River). In other studies, tributaries have been reported to have fish community structures distinct from those of the main stem [61,62]. For instance, Tsuboi et al. [62] reported that tributaries support landlocked salmonids better than the mainstream because of their physically and hydraulically complex instream habitats. The tributary analysed in this study exhibited widths and lengths similar to those of the mainstream and likely had a comparable ecosystem. Nevertheless, the community structure in winter differed between the tributary and mainstem between Sts. 4 and 5, even though they were located close together, likely indicating that this part of the river is differently utilised by the fish community during harsh seasons [63]. Alternatively, the relatively high feeding pressure from avian predators because of reduction in water levels during winter may have induced such a discrepancy [63].

For analysing the effects of anthropogenic factors, we also investigated the effects of weirs and water quality on fish communities. Small goby (*Gymnogobius urotaenia*) and Japanese gudgeon (*Pseudogobio agathonectris*) were not detected upstream of the 10 m weir in the Ikeuchi River (which has an outlet at 2.2 m), suggesting that the weir may have acted as a migration barrier for these species. *G. urotaenia* is amphidromous fish, and the juveniles exhibit a pelagic lifestyle [64]. This species has relatively low swimming ability, and it has been reported that most individuals measuring 3–4 cm were unable to pass through fishways [65]. Therefore, it is reasonable to consider that the weirs in this study may act as a barrier to their movement. *P. agathonectris* is a benthic fish that inhabits shallow areas with gentle currents [66]. Similar to *Gymnogobius urotaenia*, it may have difficulty crossing weirs. However, since many other fish species of similar size and life history were observed upstream of the weir, it remains unclear whether the weir completely restricts fish movement. Even if the weir were functioning as a complete migration barrier, the downstream population could still be supplied by individuals from the upstream population of the weir [67], resulting in the detection of both individuals and eDNA on both sides of the weir. Meanwhile, the OTU perhaps associated with Japanese crucian carps or goldenfish (*Carassius*) was detected downstream of the weir in the Isazu River but was not found upstream. This suggests that weirs may be a limiting factor for relatively large fish, such as crucian carp and *Carassius* spp.[67]. However, *Carassius* was also found upstream of the weir in the Ikeuchi River. It is possible that *Carassius* was artificially introduced upstream of

the weir. Therefore, weirs may partially play as migration barrier for the species inhabiting the surrounding areas, but the possibility that fish are moving between the upstream and downstream of the weir cannot be ruled out.

In terms of water quality, another anthropogenic factor, the pollution index was below the environmental quality standards of the Ministry of the Environment of Japan [68], making it a relatively less polluted river, at least based on the criteria of $NO_3$-N and $NO_2$-N levels < 10 mg/L; however, the criterion for $PO_4$-P pollution has not been defined. The concentrations of these chemicals correlated with the distance from the river mouth. In fact, the population is more concentrated near the river mouth than in the upstream areas of the Isazu River. Therefore, it cannot be concluded that these factors directly affect the fish community, as the community structure was also influenced by the distance from the river mouth, making it difficult to disentangle anthropogenic effects from natural spatial gradients. Meanwhile, there are rice fields near the Isazu River, and the dissolved substances investigated in this study may have originated from pesticides and fertilisers [69]. In general, water pollution is higher downstream, whereas the results obtained in our study showed that $NO_3$-N concentrations were higher upstream. Considering that St. 12 is located upstream of a nearby rice field, the source is unlikely to be agriculture. Healthy climax forests are often saturated with $NO_3$-N and, thus, provide a consistent amount of these chemicals, which are then used in primary production in rivers [70]. Alternatively, high $NO_3$-N concentrations in upstream may originate from yellow sand transported from the Eurasian continent via wind [71]. Because the upstream region contains a smaller water volume than downstream areas, the $NO_3$-N concentration may appear higher due to reduced dilution. Although the effect of anthropogenic factor on fish community is still unclear, it is important to comprehensively evaluate the effects of both the natural environment and anthropogenic activities on the Isazu River to preserve its water quality and monitor fish communities.

We noted the seasonal migration of several fishery resources. The provision of fishery resources is a major ecosystem service [28]. Ayu sweetfish (*P. altivelis altivelis*), an amphidromous fish, spawns in rivers in fall, with juveniles returning from the ocean in spring and growing from spring to fall [72]. These results indicate that the eDNA patterns reflect the ecology of fish species at different sites. They were abundant at the mouth of both rivers in early spring and in the middle and upper reaches of the rivers in summer and fall and were absent in winter. Another important fishery resource, the Japanese eel (*A. japonica*), was also detected in this river system. They migrate downriver in fall and winter, spawn near the Mariana Trench, and the juveniles return to the rivers in early spring [73,74]. The detection of the commercially important but endangered *A. japonica* illustrates the importance of the local river in fishery management. However, the eDNA concentration of *A. japonica* was low in winter (2 copies/µL), and species detected at low eDNA concentrations lack sufficient evidence of actual presence, requiring physical capture or visual confirmation. Although it is not clear whether the detection of eDNA reflects the migration of these two fish species, particularly the *A. japonica*, the seasonal eDNA patterns probably follow a known migration pattern. This assumption suggests that eDNA can be used to help manage fishery resources [39,75]. Seasonal eDNA patterns highlighted the important role played by local rivers in supporting major fishery resources.

In addition, we found the eDNA of endangered species. The conservation of endangered species is important for the sustainability of biodiversity and ecosystem services [76]. Several endangered species were detected only in summer, likely because of the generally high detection of eDNA during the warm season. Nevertheless, several endangered species were detected throughout the year (*A. japonica, L. torrentis*, and *L. reinii*). Although ecological information on Japanese eels is comparatively abundant because they are also a major fishery resource, there is insufficient information on *L. torrentis* and *L. reinii*. The mating season of *L. torrentis* is considered to be from May to July, with water temperatures ranging from 13 to 18°C [77]. The reproductive season of *L. reinii* is also believed to be from May to July, similar to that reported for *L. torrentis* [78]. Although regional and chronological differences must be considered, detecting the eDNA of these fish species at multiple sites in spring may reflect their mating activity. During the mating season, sperm are released and flow downstream and may be detected as eDNA [79]. The detection of the eDNA of these endangered fish

species across the spatiotemporal span provides a better understanding of their ecology and provides valuable insights for guiding efforts aimed at their conservation.

In conclusion, we revealed that fish community structures show spatiotemporal differences in the Isazu River Basin in temperate Japan. The fish community structure varied from the river mouth to upstream, regardless of the tributary or main stem but was dependent on season. The weirs did not seem to be the main limiting factor for fish migration, as most detected OTUs were present both above and below the weirs, and the population structure did not differ upstream and downstream of the weirs. Several fishery resources, among which at least *P. altivelis altivelis*, may exhibit known seasonal migration patterns. In addition, we detected several endangered species and provided their ecological information. However, this study has limitations: (1) eDNA samplings were conducted at only four time points during the year, (2) the impact of artificial structures other than the two weirs was not considered, and (3) other water quality parameters related to pollution, such as pH and dissolved oxygen levels, were not measured. More frequent monitoring that considers various factors is essential for evaluating whole-river connectivity and ecosystem health. Future long-term and detailed studies will provide a relatively clear picture of transitions in fish community structures and the impact of natural and anthropogenic factors on ecosystem services.

## Supporting information

**S1 Table. Information on sampling location.**
(DOCX)

**S2 Table. The DNA concentrations (ng/µL) after purification of the first PCR.**
(XLSX)

**S3 Table. Detected reads number of the internal standard DNA in each season and site.**
(XLSX)

**S4 Table. Summary of species detected across all seasons and sites, along with their eDNA concentrations (copies/L).**
(XLSX)

**S5 Fig. NMDS plot with vectors of environmental factors and underwater materials.** (a) Spring, (b) summer, (c) fall, (d) winter, and (e) all season. The distance indicates distance from the river mouth. In all season, sp, su, fa, and wi indicate spring, summer, fall, and winter, respectively.
(PDF)

**S6 Fig. The relationship between the distance from the river mouth and the concentration of underwater materials in different seasons.** (a) $NO_2$-N, (b) $NO_3$-N, and (c) $PO_4$-P. The left panels indicate the difference of these concentrations among season, and the right panels show the relationship between the distance and these concentrations.
(PDF)

## Acknowledgments

We thank Osamu Mukai at the Maizuru Fisheries Research Station for helping with the water quality analysis. We also would like to thank Editage (www.editage.jp) for English language editing.

## Author contributions

**Conceptualization:** Reiji Masuda.

**Funding acquisition:** Reiji Masuda.

**Investigation:** Takeshi Ito, Tetsu Yatsuyanagi, Tomohiro Yokobe, Misaki Shiomi, Reiji Masuda.

**Methodology:** Takeshi Ito, Tetsu Yatsuyanagi, Tomohiro Yokobe.

**Project administration:** Reiji Masuda.

**Supervision:** Reiji Masuda.

**Visualization:** Takeshi Ito.

**Writing – original draft:** Takeshi Ito.

**Writing – review & editing:** Tetsu Yatsuyanagi, Tomohiro Yokobe, Misaki Shiomi, Reiji Masuda.

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
