## [Decision Letter · Decision Letter 0]

PONE-D-25-12060Seasonal changes in the structure of river fish communities in temperate Japan depicted using quantitative eDNA metabarcodingPLOS ONE

Dear Dr. Ito,

Thank you for submitting your manuscript to PLOS ONE. After careful consideration, we feel that it has merit but does not fully meet PLOS ONE’s publication criteria as it currently stands. Therefore, we invite you to submit a revised version of the manuscript that addresses the points raised during the review process.

We look forward to receiving your revised manuscript.

Kind regards,

Qing Wang

Academic Editor

PLOS ONE

Journal Requirements:

2. Thank you for stating the following financial disclosure: [This project was funded by the AEON Environmental Foundation to RM and Japan Society for the Promotion of Science KAKENHI (grant number: 19H05641 to RM).]. 

Reviewers' comments:

Reviewer's Responses to Questions

**Comments to the Author**

1. Is the manuscript technically sound, and do the data support the conclusions?

Reviewer #1: Yes

Reviewer #2: Yes

2. Has the statistical analysis been performed appropriately and rigorously? 

Reviewer #1: Yes

Reviewer #2: Yes

3. Have the authors made all data underlying the findings in their manuscript fully available?

Reviewer #1: No

Reviewer #2: Yes

4. Is the manuscript presented in an intelligible fashion and written in standard English?

Reviewer #1: Yes

Reviewer #2: Yes

5. Review Comments to the Author

Reviewer #1: The authors detected fish species in Isazu river and its tributary Ikeuchi river using quantitative environmental DNA metabarcoding, and detected 78 species (or operational taxonomic units) across the seasons. The found that the fish community structure varied spatially and temporally. I believe that this work appeals to a broad scientific community particularly those ae interested in fish biology and coastal/marine science. I have some minor suggestions that might improve the presentation of the manuscript.

Minor comments:

L19. This study does not go beyond fish species detection, and does not study ecosystem services per se.

L267. Comprised.

L294. Space-time -> Spatio-temporal

L298. The spring and the fall; delete ‘pair’

L305. The authors should present degree of freedom of F statistics in Table1

L322. Independent -> dissimilar?

L461. Nitrogen deposition from yellow sand impact different fish species equally. I’m not sure why does this factor come into play.

Figure 3. Distance dependence. It is known that Shannon diversity index recovers its doubling property and the exponentiated Shannon index can thus be better linked with distance from the river mouth. I suggest that the current version s fine, but merits redrawing the figure using exponentiated index; then the authors fid exponential decay more clearly.

See the Equation (B3) of the following article by Roswell et al. 2021.

Reference: https://nsojournals.onlinelibrary.wiley.com/doi/10.1111/oik.07202

Reviewer #2: In the manuscript titled “Seasonal changes in the structure of river fish communities in temperate Japan depicted using quantitative eDNA metabarcoding,” the authors have employed an eDNA metabarcoding approach to examine spatial and seasonal variations in fish communities within the Isazu River.

The manuscript is well-written, clearly organized, and easy to follow. The analyses appear sound, with no major methodological flaws identified. The conclusions are well supported by the data presented. However, one significant concern is the limited sampling effort used to assess seasonal changes in fish diversity. The authors collected samples only once per season, which is insufficient to draw robust conclusions about seasonal variation in fish communities.

Another concern pertains to the quality of the figures. In several instances, the text within the figures is blurry and difficult to read. I recommend that the authors improve the resolution and overall quality of the figures.

Minor Comments:

• Line 85: Please delete the word “so.”

• Line 146: Likely a typo—please correct “10 mL” to “10 μL.”

• Line 159: Include the DNA concentration in ng/μL. Listing only the volume (1 μL) is not informative.

• Line 193: Clarify how the value of 200 was determined to convert copy number/μL to copy number/L.

• Line 279: Replace “Benn” with “Venn.”

• Line 293: Remove the space between “H′” and “during.”

• Line 310: The “4” in “PO4-P” should be subscripted.

• Line 337: Change “2 copy/L” to “2 copies/L.”

• Lines 342–343: The use of two-letter abbreviations for Le. torrentis and Li. reinii seems unnecessary, as there is no apparent ambiguity in these taxa. Please explain the rationale or correct it consistently throughout the manuscript.

• Line 355: Add the word “in” before “the.”

• Line 410: The phrase “in other studies” is used, but only one reference is cited. Please include additional references.

• Line 426: Remove the extra comma.

• Line 437: “The” should begin with a lowercase “t.”

• Line 462: Clarify what is meant by “continent” in the phrase “yellow sand transported from the continent.”

6. PLOS authors have the option to publish the peer review history of their article (what does this mean? ). If published, this will include your full peer review and any attached files.

**Do you want your identity to be public for this peer review?** For information about this choice, including consent withdrawal, please see our Privacy Policy .

Reviewer #1: No

Reviewer #2: **Yes: ** Girish Kumar

---

## [Author Response · Author response to Decision Letter 1]

26 May 2025

REPLY: We are very glad to receive a positive evaluation of our manuscript. We have revised the manuscript based on Reviewers’ comments.

Reviewer #1: The authors detected fish species in Isazu river and its tributary Ikeuchi river using quantitative environmental DNA metabarcoding, and detected 78 species (or operational taxonomic units) across the seasons. The found that the fish community structure varied spatially and temporally. I believe that this work appeals to a broad scientific community particularly those ae interested in fish biology and coastal/marine science. I have some minor suggestions that might improve the presentation of the manuscript.

REPLY: Thank you for your constructive comments. I have revised the manuscript based on your suggestions.

Minor comments:

L19. This study does not go beyond fish species detection, and does not study ecosystem services per se

REPLY: .We have deleted the sentence. Please see L18.

L267. Comprised.

REPLY: We have corrected the sentence. Please see L269.

L294. Space-time -> Spatio-temporal

REPLY: We have corrected the sentence. Please see L298.

L298. The spring and the fall; delete ‘pair’

REPLY: We have deleted it. Please see L302.

L305. The authors should present degree of freedom of F statistics in Table1

REPLY: We have added between-group degrees of freedom and residual degree of freedom in Table 1and main text. Please see L300, L2303, and Table 1 in L308.

L322. Independent -> dissimilar?

REPLY: We have corrected the sentence to make it clear. Please see L326-330.

L461. Nitrogen deposition from yellow sand impact different fish species equally. I’m not sure why does this factor come into play.

REPLY: We have revised the text to clarify why NO₃-N concentrations were high in the upstream region. While the direct effect of this pollution on the fish community remains unclear, we believe that considering the potential causes of increased NO₃-N concentration is important for understanding environmental conditions in the river. Please see L467-471.

Figure 3. Distance dependence. It is known that Shannon diversity index recovers its doubling property and the exponentiated Shannon index can thus be better linked with distance from the river mouth. I suggest that the current version s fine, but merits redrawing the figure using exponentiated index; then the authors fid exponential decay more clearly.

See the Equation (B3) of the following article by Roswell et al. 2021.

Reference: https://nsojournals.onlinelibrary.wiley.com/doi/10.1111/oik.07202

REPLY: Thank you for your recommendation. We have exponentiated Shannon index and used it. Please see L211-219 and. Figure 3.

Reviewer #2: In the manuscript titled “Seasonal changes in the structure of river fish communities in temperate Japan depicted using quantitative eDNA metabarcoding,” the authors have employed an eDNA metabarcoding approach to examine spatial and seasonal variations in fish communities within the Isazu River.

The manuscript is well-written, clearly organized, and easy to follow. The analyses appear sound, with no major methodological flaws identified. The conclusions are well supported by the data presented. However, one significant concern is the limited sampling effort used to assess seasonal changes in fish diversity. The authors collected samples only once per season, which is insufficient to draw robust conclusions about seasonal variation in fish communities.

Another concern pertains to the quality of the figures. In several instances, the text within the figures is blurry and difficult to read. I recommend that the authors improve the resolution and overall quality of the figures.

REPLY: We agree with your point regarding the limited sampling effort. Although we believe that meaningful insights can still be drawn from small sample sizes, we recognize that the results are preliminary and have made efforts to interpret them with appropriate caution.

Regarding the resolution of the figures, we apologize for any inconvenience. The resolution appears fine on our end, so it is possible that the issue may be related to how the figures were displayed in the review system. If that is the case, downloading the figures using the button at the upper right corner of each figure page might help improve visibility. However, if the problem persists, we would be happy to provide higher-resolution versions.

Minor Comments:

• Line 85: Please delete the word “so.”

REPLY: We have deleted it. Please see L81.

• Line 146: Likely a typo—please correct “10 mL” to “10 μL.”

REPLY: We have corrected it. Please see L143.

• Line 159: Include the DNA concentration in ng/μL. Listing only the volume (1 μL) is not informative.

REPLY: We did not measure the DNA concentration of the PCR template prior to the first PCR. Instead, we have summarized the concentration of the first PCR products in the Supporting Information. Please see L163-165 and S2 Table.

• Line 193: Clarify how the value of 200 was determined to convert copy number/μL to copy number/L.

REPLY: We have clarified in the manuscript how the value of 200 was derived to convert the copy number per μL to the copy number per L. Please see L192-194.

• Line 279: Replace “Benn” with “Venn.”

REPLY: We have replaced it. Please see L282.

• Line 293: Remove the space between “H′” and “during.”

REPLY: We have removed it. Please see L297.

• Line 310: The “4” in “PO4-P” should be subscripted.

REPLY: We have corrected it. Please see L316.

• Line 337: Change “2 copy/L” to “2 copies/L.”

REPLY: We corrected them throughout the entire manuscript. Please see L24, 260,261, 343, 356, and 487.

• Lines 342–343: The use of two-letter abbreviations for Le. torrentis and Li. reinii seems unnecessary, as there is no apparent ambiguity in these taxa. Please explain the rationale or correct it consistently throughout the manuscript.

REPLY: We have corrected them throughout the entire manuscript. Please see L348, 349, 351, 498,500, 501, and 503.

• Line 355: Add the word “in” before “the.”

REPLY: We have added it. Please see L361.

• Line 410: The phrase “in other studies” is used, but only one reference is cited. Please include additional references.

REPLY: We have added a reference. Please see L417.

• Line 426: Remove the extra comma.

REPLY: We have removed unnecessary space. Please see L432.

• Line 437: “The” should begin with a lowercase “t.”

REPLY: We have corrected it Please see L443.

• Line 462: Clarify what is meant by “continent” in the phrase “yellow sand transported from the continent.”

REPLY: In this context, “continent” refers specifically to the Eurasian continent, particularly the arid and semi-arid regions of China and Mongolia, which are the major source areas of yellow sand (Asian dust) transported to Japan. To improve clarity, we have revised the sentence accordingly. Please see L 469-472.

---

## [Decision Letter · Decision Letter 1]

Seasonal changes in the structure of river fish communities in temperate Japan depicted using quantitative eDNA metabarcoding

PONE-D-25-12060R1

Dear Dr. Ito,

We’re pleased to inform you that your manuscript has been judged scientifically suitable for publication and will be formally accepted for publication once it meets all outstanding technical requirements.

Kind regards,

Qing Wang

Academic Editor

PLOS ONE

Additional Editor Comments (optional):

Reviewers' comments:

Reviewer's Responses to Questions

**Comments to the Author**

1. If the authors have adequately addressed your comments raised in a previous round of review and you feel that this manuscript is now acceptable for publication, you may indicate that here to bypass the “Comments to the Author” section, enter your conflict of interest statement in the “Confidential to Editor” section, and submit your "Accept" recommendation.

Reviewer #2: All comments have been addressed

2. Is the manuscript technically sound, and do the data support the conclusions?

Reviewer #2: Yes

3. Has the statistical analysis been performed appropriately and rigorously? 

Reviewer #2: Yes

4. Have the authors made all data underlying the findings in their manuscript fully available?

Reviewer #2: Yes

5. Is the manuscript presented in an intelligible fashion and written in standard English?

Reviewer #2: Yes

6. Review Comments to the Author

Reviewer #2: (No Response)

7. PLOS authors have the option to publish the peer review history of their article (what does this mean? ). If published, this will include your full peer review and any attached files.

**Do you want your identity to be public for this peer review?** For information about this choice, including consent withdrawal, please see our Privacy Policy .

Reviewer #2: No

---

## [Editor Report · Acceptance letter]

PONE-D-25-12060R1

PLOS ONE

Dear Dr. Ito,

I'm pleased to inform you that your manuscript has been deemed suitable for publication in PLOS ONE. Congratulations! Your manuscript is now being handed over to our production team.

Kind regards,

on behalf of

Dr. Qing Wang

Academic Editor

PLOS ONE